

# Seasonal variation of atmospheric particle number concentrations, new particle formation and atmospheric oxidation capacity at the high Arctic site Villum Research Station, Station Nord

Q.T. Nguyen[1,2,3], M. Glasius[2,4], L.L. Sørensen[1,6], B. Jensen[1], H. Skov[1,5,6], W. Birmili[7], A. Wiedensohler[7], A. Kristensson[8], J.K. Nøjgaard[1] and A. Massling[1,6]

[1] Department of Environmental Science, Aarhus University, 4000 Roskilde, Denmark

[2] Department of Chemistry, Aarhus University, 8000 Aarhus, Denmark

[3] Department of Engineering, Aarhus University, 8200 Aarhus, Denmark

[4] Interdisciplinary Nanoscience Center (iNANO), Aarhus University, 8000 Aarhus, Denmark

[5] Institute of Chemical Engineering and Biotechnology and Environmental Technology, University of Southern Denmark, 5230 Odense, Denmark

[6] Arctic Research Centre, Aarhus University, 8000 Aarhus, Denmark

[7] Leibniz Institute for Tropospheric Research, 04318 Leipzig, Germany

[8] Department of Physics, Lund University, Sweden

Correspondence to: Q.T. Nguyen (quynh@eng.au.dk)

## Abstract

This work presents an analysis of the physical properties of sub-micrometer aerosol particles measured at the high Arctic site Villum Research Station, Station Nord (VRS), northeast Greenland between July 2010 and February 2013. The study focus on particle number concentrations, particle number size distributions, the occurrence of new particle formation (NPF) events and their seasonality in the high Arctic, where observations and characterization of such aerosol particle properties and corresponding events are rare and understanding of related processes is lacking.





A clear accumulation mode was observed during the darker months from October until mid-May,
which became considerably more pronounced during the prominent Arctic haze months from March
to mid-May. In contrast, only nucleation and Aitken-mode particles were observed during the
summer months. Analysis of wind direction and wind speed indicated possible contributions of
marine sources from the easterly side of the station to the observed summertime particle number
concentrations, while southwesterly to westerly winds dominated during the darker months. NPF
events lasting from hours to days were mostly observed from June until August, with fewer events
observed during the months with less sunlight March, April, September, and October. It was
observed that ozone ($O_3$) is likely to play an important role in the formation and growth of new
particles at the site during summertime. Calculations of air-mass back trajectories using the Hybrid
Single Particle Lagrangian Integrated Trajectory (HYSPLIT) model for the NPF event days
suggested that the events possibly originated from other places and transported together with $O_3$ in
air parcels from different heights of the boundary layer down to the station at ground level. A map
of event occurrence probability was computed, indicating that southerly air masses from over the
Greenland Sea were more likely linked to those events.

## 1. Introduction

Climate change driven by anthropogenic greenhouse gas emissions is a global challenge. In the
Arctic, the warming climate has already led to an earlier onset of spring-ice melt, later freeze-up
and decreasing sea-ice extent (Zwally et al., 2002; Markus et al., 2009; Stroeve et al., 2012). The
reduction of the Earth's albedo due to ice loss subsequently impacts the radiative balance of the
Earth through a positive feedback, leading to further warming. As a result, the Arctic has been
considered a manifestation of global warming with the rate of temperature increase in the region
being twice as high as the rest of the world (IPCC, 2013; ACIA, 2005), up to 8 - 9 °C along the east
coast of Greenland (Stendel et al., 2008). In addition to long-lived greenhouse gases, short-lived
climate forcers including tropospheric ozone, aerosols and black carbon also play a significant role
affecting the radiative balance in the Arctic (Quinn et al., 2008; Bond et al., 2013; IPCC, 2013).
Aerosol particles influence the radiative balance in the Arctic in many ways, through their ability to
absorb and scatter incoming solar radiation or by acting as cloud condensation nuclei to form cloud
and fog droplets. The presence of low level liquid clouds above bright ice- and snow-covered
surfaces in the Arctic could lead to increasing near-surface temperature as opposed to a cooling
effect observed in most other global regions (Shupe and Intrieri, 2004; Bennartz et al., 2013),
though the effect is probably small (AMAP, 2011). At the same time, deposition of black carbon on
Arctic snow- and ice-covered surfaces accelerates surface heating and ice melting in early spring





(Hansen and Nazarenko, 2004; Flanner et al., 2007; Flanner et al., 2009). It is thus crucial to
investigate the dynamics of atmospheric aerosol particles observed in the Arctic (involving the
formation, concentration, physico-chemical properties, temporal variability and transport) to
understand their direct and indirect effects on the radiation budget.
It is well known that during each winter extending into spring, Arctic aerosol particles containing
mineral dust, black carbon, heavy metals, elements, sulfur and nitrogen compounds are detected in
elevated concentrations. This has been attributed to the annually recurring Arctic haze phenomenon,
which is related to distant latitude anthropogenic pollution (Li and Barrie, 1993; Quinn et al., 2002;
Ström et al., 2003; Heidam et al., 2004; Heidam et al., 1999; Nguyen et al., 2013). The focus was
thus on long-range transported aerosols, which are expected to be aged due to the long transport
distance from mid-latitude source regions.
A number of studies have reported in-situ formation of new aerosol particles in the Arctic, which
mostly involved new particle formation in the Arctic boundary layer. The first observations of the
occurrence of an ultrafine particle mode (< 20 nm) in the Arctic marine boundary layer during
summer and autumn were reported by Wiedensohler et al. (1996) and Covert et al. (1996).
Observations of small aerosol particles during the summer period have also been reported at the
Zeppelin mountain site, Svalbard (11.9°E, 78.9°N, 478 m a.s.l.) within the Arctic boundary layer
(Ström et al., 2003; Tunved et al., 2013). The current understanding on mechanisms of new particle
formation in the marine boundary layer over the Arctic Ocean is unclear, due to the low
concentration of nucleating agents such as sulfuric acid in the marine boundary layer (Pirjola et al.,
2000; Karl et al., 2012), in addition to the limited number of observational data. Growth of ultrafine
particles has been observed at Summit, Greenland (38.4°W, 72.6°N, 3200 m a.s.l.) (Ziemba et al.,
2010). Quinn et al. (2002) also found an increase in particle number concentrations during the
summer months at Barrow, Alaska (156.6°W, 71.3°N, 8 m a.s.l.), which was attributed to the
formation of smaller particles. A correlation between summertime particle number concentrations
and the biogenic production of methane sulfonate (MSA⁻) was shown, indicating that the production
of summertime particles may be associated with biogenic sulfur (Quinn et al., 2002). Similar
finding has been recently reported by Leaitch et al. (2013) based on observations from Alert,
Nunavut. Heintzenberg et al. (2015) observed newly formed small aerosol particles during several
cruises to the summer central Arctic Ocean and suggested that they could originate from around the
Arctic region, more specifically related to air masses passing by open waters prior to the
observation point.
Meanwhile, source regions of aerosol particles in the Arctic could be very different (Hirdman et al.,
2010). Barrow is mostly influenced by North America and Arctic basin with some Russian and





Siberian sources (Quinn et al., 2002). Summit, which is located above the planetary boundary layer,
receives frequent long-range transported pollution from North America and extensively from
Eurasia during wintertime (Kahl et al., 1997; Hirdman et al., 2010). The mountainous site Zeppelin
(Tunved et al., 2013) and the ground level site VRS (16°40'W, 81°36'N, 30 m a.s.l.) (Heidam et al.,
2004; Nguyen et al., 2013) both receive long range transported pollution predominantly from
Eurasia during winter and spring. Zeppelin is often located south of the Polar Front receiving
transport from the Atlantic Ocean during summer (Tunved et al., 2013). Svalbard is also influenced
by the Gulf Stream (Pnyushkov et al., 2013) and surrounded by open sea during summertime. VRS
is influenced by the ice stream from the Arctic Ocean along the east coast of Greenland (Stendel et
al., 2008; Kwok, 2009) and surrounded by multi-year sea ice, with limited first-year ice along the
coast. Such differences could have considerable impacts on NPF events and also aerosol particle
properties, which requires investigations at high spatial resolution in the Arctic.
VRS, Station Nord is a unique coastal station located close to sea level, representing the conditions
of the high Arctic throughout the year. Until date, there is only one observation and characterization
of NPF events at Alert, Nunavut (Leaitch et al., 2013), while understanding of particle size
distribution, seasonality as well as related mechanisms and processes of NPF events are lacking
from such a high Arctic site.
This study aims to characterize the formation, concentration, physical properties and seasonality of
atmospheric aerosols based on particle number size distributions at VRS. The occurrence of NPF
events was investigated in details. The events were classified and analyzed together with ozone ($O_3$)
and nitrogen oxides ($NO_x = NO + NO_2$). Wind direction and wind speed were analyzed to
investigate the impacts of source regions on the observed seasonality of particle number size
distribution. The source regions of new particle formation were mapped based on calculations of air
mass back trajectories using the HYSPLIT model during event days and non-event days. A
probability map for NPF event occurrence was computed.
**2. Methods**
**2.1. Measurement site**
Aerosol particles and trace gases were measured at the measurement site "Flyger's Hut", VRS,
Station Nord in northeast Greenland (81°36'N, 16°40'W, 30 m a.s.l.). The site is located on a small
peninsula (Princess Ingeborgs Peninsula) at approximately 2.5 km southeast of a small Danish
military base housing a crew of five soldiers (**Fig. 1**). Electricity at "Flyger's Hut" is supplied from
a local JET A-1 fuel generator located inside the military base. The remote location of the station
implies a minor, though unavoidable, contribution of local anthropogenic pollution originating from



the military camp. The station is surrounded by multi-year sea ice, with limited bare ground
occasionally and limited first-year ice along the coast of Greenland during the summer months. At
VRS, Station Nord, polar sunrise is observed in the end of February, while polar day prevails from
mid-April to the beginning of September and polar night prevails from mid-October to the end of
February.
**2.2. Instrumentation**
**2.2.1. Mobility Particle Size Spectrometer**
Measurement of particle number size distributions at Station Nord was initiated in July 2010 using a
TROPOS-type Mobility Particle Size Spectrometer as described in Wiedensohler et al. (2012).
Briefly, the instrument consists of a medium Vienna-type Differential Mobility Analyzer (DMA)
followed by a butanol-based Condensation Particle Counter (CPC 3772 by TSI Inc., Shoreview,
USA). The DMA design is described in Winklmayr et al. (1991). The system is operated at 1 l min$^{-1}$
aerosol flow rate and 5 l min$^{-1}$ sheath air flow rate. The DMA sheath flow is circulated in closed
loop, facilitated by a regulated air blower. This technical setup allows measurements across a
particle size range from 10 to 900 nm in diameter. The time resolution of the instrument is 5 min,
including up-scan and down-scan.
The instrument was specifically designed to allow long-term operation with minimum maintenance
as follows. The DMA sheath air flow rate was continuously measured using a calibrated mass flow
sensor. The DMA aerosol flow rate was monitored by a pressure drop measurement over a
calibrated capillary. A computer-based control program adjusted the sheath air flow rate after each
measurement of the particle number size distribution. Systematic deviations in the sample flow rate,
which was controlled by a critical orifice in the CPC were monitored and corrected for in the
successive size distribution evaluation. Additionally, absolute pressure was measured at the inlet of
the system to detect any substantial technical problems such as clogging of the inlet. Temperature
and relative humidity (RH) were monitored at several positions inside the instrument. The RH
inside the DMA is the most critical parameter, since excessive moisture would allow particles to
grow much beyond their nominal dry diameter. At VRS, Station Nord, RH is usually not a critical
issue, as the climate is cold and arid with low humidity most of the year. The temperature in the
laboratory is mostly considerably higher than outdoor temperature, implying that substantial drying
of the aerosol is not needed most of the time during sample intake into the laboratory.
**2.2.2. Data processing**





The raw particle electrical mobility distributions collected by the mobility particle size spectrometer
were processed by a linear inversion algorithm presented in Pfeifer et al. (2014).
As a first part of quality control, any data associated with DMA excess air RH above 50 % and
sheath air temperature above 30 °C were excluded from further data analysis, as recommended by
ACTRIS and WMO-GAW (http://www.wmo-gaw-wcc-aerosol-physics.org/recommen-
dations.html). These incidents were only observed on a few days during the study period.
Subsequently, daily particle number size distributions were plotted to inspect any sudden increase in
the particle number concentration above the background. If short-lived particle number
concentration peaked without any detectable particle growth coincided with similar peaks of $NO_x$,
they were interpreted as local pollution events and excluded from the data set. These local pollution
events were observed throughout the year at the station. **Fig. 2** shows the extent of data coverage
over the study period. Gaps in the data set (most notably in 2011) were due to excluded data with
flow uncertainties. 2012 was the year with the best data coverage, with the lowest percentage of ca.
78 % in March while exceeding 90 % in most other months. The year 2012 was therefore chosen to
examine the seasonality of Arctic aerosols in details.
**2.2.2. Gas phase and meteorological parameters**
$O_3$ was measured using an API photometric $O_3$ analyzer (M400). The results were averaged to a
time resolution of 30 min. The detection limit was 1 ppbv with an uncertainty of 3 % and 6 % for
measured concentrations above and below 10 ppbv, respectively. The uncertainties were calculated
at 95 % confidence interval.
$NO_x$ was averaged to a time resolution of 30 min (Teledyne API M200AU, San Diego, CA) with a
precision of 5 % and a detection limit of 150 ppt. The calibration was checked weekly using 345
ppb NO span gas while zero gas was added each 25 hour. $NO_x$ was sampled at a flow rate of 1 l
$min^{-1}$. Coverage of $O_3$ and $NO_x$ data in this study are indicated as the corresponding blue and red
line in **Fig. 2**.
Wind speed and wind direction data were obtained from a sonic anemometer (METEK, USA-1,
heated) for the period from April 2011 to April 2013. In winter periods fewer data were obtained
due to frost on the anemometer when temperature was below approximately -35 °C.
**2.3.  Classification of new particle formation events**





NPF events were identified and classified following a scheme adapted from Dal Maso et al. (2005).
A brief description is given here.
A plot was compiled for each day with available particle number size distribution data, plotting the
particle diameter on the y-axis, time of the day (from midnight to midnight) on the x-axis, with the
particle number concentration in each size interval displayed as a contour plot. A panel of three
persons performed visual inspection, identification and classification of data to avoid subjective
bias. In order to be classified as an event day, the occurrence of a new particle mode below 20 nm
with concentrations substantially higher than during the previous hours must be observed. If a clear
diameter growth of newly formed particles could be traced for several hours, that specific day
would be classified as a class I event day. If the growth of newly formed particles was not
continuous over several hours, that specific day would be classified as a class II event day. The
identified NPF events at Station Nord typically lasted from hours to days. In case of a multi-day
event, only the first day, during which the event onset was identified, was counted as an event day.
The panel must agree on all classifications, otherwise the specific day would be classified as an
undefined event. Other options for classifications are non-event day or bad data in case of missing
data or observed instrumental problems.

**3. Results and Discussion**

This section presents the observed overall seasonality of particle number size distributions
measured at VRS, Station Nord during the time period from July 2010 to February 2013, with an
analysis of NPF event cases together with the atmospheric oxidation capacity at the station.
Analysis of local wind speed, wind direction and air mass back trajectories was used to support the
interpretation of the seasonality of particle number size distributions and the dynamics of NPF
events.

**3.1. Particle number size distributions and seasonality**

**3.1.1. Overview**

A clear seasonality of particle number size distributions was observed during 2012 (**Fig. 3-4**). A
persistent accumulation mode appeared in the end of September, which became more prominent in
the end of February lasting until mid-May. The Arctic summer (June - August) was coupled with a
higher abundance of nucleation mode and Aitken mode aerosol particles and a very low abundance
of accumulation mode particles (Table 1). The small particles were also observed to a lesser extent



in September and only during one episode in mid-October. This observation of strong seasonality
was supported by observations from the available scattered data in the other years 2010, 2011 and
2013. The elevated concentrations of accumulation mode particles observed in this study generally
followed the varying pattern of aged total suspended particles during the Arctic haze period
previously reported at VRS, Station Nord (Heidam et al., 2004; Nguyen et al., 2013) and other
Arctic stations (Quinn et al., 2002; Ström et al., 2003). It should also be noted that the sun rises in
the end of February at Station Nord, so the period thereafter is affected by photochemical processes.
Observations of smaller particles during this period were in accordance with previous studies in the
Arctic (Ström et al., 2003; Tunved et al., 2013; Wiedensohler et al., 1996; Covert et al., 1996;
Quinn et al., 2002; Heintzenberg et al., 2015; Leaitch et al., 2013). During this period, the Arctic is
considerably cleaner with respect to long-range transport of atmospheric pollutants and
characterized by constant daylight.
**3.1.2. Statistics of the particle number size distribution**
**Fig. 4** and **Table 1** describe detailed statistics of the particle number size distributions measured at
the site, especially regarding the prominent accumulation mode during February - May and the
prominent nucleation/Aitken mode during June - August. **Table 2** provides detailed median and
average particle number concentration (N), particle volume concentration (V) and particle mass
concentration (M) values calculated using the particle number size distributions at VRS, Station
Nord during 2012. Higher values of median or average N were observed from April to September.
During this period, largest discrepancies between the median and the average values were also
found, especially during June (Median N = 137 particles cm$^{-3}$, Average N = 277 particles cm$^{-3}$) and
August (Median N = 227 particles cm$^{-3}$, Average N = 313 particles cm$^{-3}$). This was attributed to the
occurrence of intense NPF events during these months (**Fig. 3**), skewing the average N towards
higher values compared to median N. June and August also showed highest average N in 2012,
followed by May, April and July, whereas the months with the lowest average N were October,
November and December. Since nucleation mode particles were almost absent in April and
relatively minor in May, their corresponding high median or average N values observed were
attributed to the elevated presence of the pronounced accumulation mode during these two months
(**Fig. 3**).
Newly formed particles are usually high in number and therewith significantly influence the total
number concentration N as discussed above; however they do not contribute considerably to the





total particle volume concentration V. As a result, June and August were among the months with
the lowest median or average V together with other sunlit months July and September (**Table 2**). In
contrast, the highest median and average V were observed during the most prominent haze months
March - May. Simple log-normal fitting applied to the accumulation mode observed in the monthly
particle number size distributions in 2012 revealed a geometrical mean diameter of  approximately
170 nm during the winter and spring months (**Table 1**). This indicates that the particles can
originate from distant locations due to their longer lifetimes determined by their size (Massling et
al., 2015).
The total particle mass concentrations M were derived directly from the total particle volume
concentration V, assuming a density of 1.4 g cm$^{-3}$ and particle sphericity. Average monthly
estimates of M ranged from 0.21 µg m$^{-3}$ (June) to 1.58 µg m$^{-3}$ (March) (**Table 2**).
Similar distribution of the major modes was also observed at the Zeppelin mountain site by Tunved
et al. (2013). However, the nucleation mode - Aitken mode observed during the summer months
seemed considerably more pronounced at VRS, Station Nord compared to Zeppelin. This indicates
higher number concentrations of smaller particles at Station Nord, which were visible until October
(**Fig. 3-4**). In regards of the total particle mass concentration, Tunved et al. (2013) reported summer
M mostly below 0.2 µg m$^{-3}$ and higher M below 0.8 µg m$^{-3}$ observed at Zeppelin during the
prominent haze months March - April (with an assumed lower density of 1 g cm$^{-3}$). Clearly, the
particle mass concentration at Villum Research Station, VRS, Station Nord seemed comparable
during summer while showing higher concentrations during the Arctic haze months compared to
Zeppelin with different assumed particle densities already accounted for. This difference between
the two sites could be partially attributed to their different locations as discussed above. In addition,
the study periods and lengths of the studies were also different, as the Zeppelin data was averaged
for March 2000 - March 2010 whereas the descriptive distribution statistics in this work was
derived solely from data in 2012. Nevertheless, similar observations at both stations show the
consistent and predictable annual behavior of the particle number size distributions in the Arctic.
**3.1.3. Impacts of seasonal wind pattern**
Analysis of wind direction and wind speed was performed to investigate the impacts of wind pattern
on the particle number size distributions at the station. **Fig. 5** demonstrates monthly wind roses
during 2012, where two distinct patterns could be identified during the darker (September - April)
and the summer (June - August) period. The early haze months (January and February) and the





prominent haze months (March and April) showed prevailing wind arriving from the southwesterly
to westerly direction. During May, some northerly wind was observed while the frequency of
southwesterly wind seemed to decrease. During the summer period (June - August), when smaller
and freshly formed particles were observed, easterly wind became more prominent, especially
during July and August. September marked a prompt change in the wind direction back to
southwesterly direction. The wind speed became higher during November - December, which is
probably due to increasing katabatic winds from the ice sheet. During the other years 2011 and
2013 (data not shown), considerably similar patterns were observed for the corresponding months.
Earlier studies on source apportionment of total suspended particles (TSP) observed during the
Arctic haze period at VRS mostly identified Siberian industries and long-range transport from mid-
latitudes as major factors (Nguyen et al., 2013; Heidam et al., 2004) . However, the wind pattern
shown here may indicate an immediate impact of the adjacent southwesterly to westerly regions
contributing to the properties of particles prior to arrival at the station.
Based on the summer wind pattern, the smaller particles observed during June - August were
probably linked to sources from the easterly side of the station, with some marine contribution.
During summer, the marine contribution from the easterly direction is possibly driven by the retreat
of sea-ice cover, which exposes areas of open waters ("open leads") and melt water on top of sea
ice to wind stress, especially along the coastal line of Greenland due to the presence of first-year-ice
in these regions. This would result in enhanced primary emissions of sea spray particles (Korhonen
et al., 2008). Surface active organic species in the ocean surface layer, which are more abundant due
to increased biological activity during summer, could also be released into the atmosphere by
bubble bursting (Middlebrook et al., 1998; Tervahattu et al., 2002) and become mixed with other
sea spray particles. It was suggested by Sellegri et al. (2006) that this could also alter the number
size distributions of particles. Another study by Karl et al. (2013) proposed that new nanoparticles
in the high Arctic could be marine granular nanogels injected into the atmosphere from evaporating
cloud droplets. Recent analysis of particle number size distributions and back trajectories during
summer cruises in the Arctic by Heintzenberg et al. (2015) also showed a high coupling of newly
formed particles and the traveling of air masses over open water. At the same time, it must be noted
that wind measurements using the sonic anemometer were confined to local observations at ground
level, which according to radio sound measurements by Batchvarova et al. (2013), do not capture
activities such as transport of air masses at higher altitudes, or transport from a broader region. The
extent of wind impacts on the particle size distributions at the station is thus not well constrained.



Previous studies reported a dependence of particle number concentrations on wind speed in the
Arctic (Leck et al., 2002) and North Atlantic (Odowd and Smith, 1993). However, in this study the
accumulation mode particles (110 - 900 nm) only showed positive correlation with wind speed
during eight out of 12 months of 2012 with a moderate Pearson correlation coefficient range of 0.05
- 0.38. The reason could be partly attributed to the larger size ranges (500 nm up to 16 μm in
diameter) measured in the other studies, which are more influenced by wind speed.
**3.2. New particle formation events**
**3.2.1. Description of exemplary NPF events**
NPF events were observed at the station during the sunlit months, especially during the summer
months June – August, though events were also identified during the months with relatively low
sunlight March and October. The onset of NPF events was observed during various hours of the day
during the summer months, in combination with very small variations in solar flux during the day.
Examples of three events were shown in **Fig. 6**. As apparent from the figure, the events showed
clear but slow growth over considerably long periods up to a few days.
**3.2.2. The role of atmospheric oxidants**
**Fig. 6** also shows an overlay of $O_3$, NO and $NO_x$ on the NPF event plots to allow analysis of the
role of atmospheric oxidants during those events.
**Ozone**
$O_3$ shows a strong seasonality in the Arctic troposphere with maximum springtime concentration
observed in the free troposphere, which is however poorly understood (Monks, 2000; Law and
Stohl, 2007). It has long been indicated that tropospheric $O_3$ in the Arctic is enriched from intruding
stratospheric air masses (Gregory et al., 1992; Gruzdev and Sitnov, 1993). A recent model study has
also suggested that summertime photochemical production of $O_3$ by $NO_x$ in the Arctic could also be
a dominant source (Walker et al., 2012). This was attributed to $NO_x$ emissions from the thermal
decomposition of the long-lived reservoir species peroxyacetyl nitrate (PAN) during summer (Fan
et al., 1994). Meanwhile, transport from mid-latitude source regions could also contribute to the $O_3$
budget in the Arctic during autumn and winter (Walker et al., 2012). Sources of $O_3$ in the Arctic
could therefore be a combination of different factors, including among others stratospheric
influence, local production and transport from mid-latitude sources. Finally, surface $O_3$ is also



depleted every spring due to reactions with Br atoms (Barrie et al., 1988; Simpson et al., 2007;
Skov et al., 2004), similar to $O_3$ depletion in the stratosphere.
In this work, $O_3$ was used as a tracer of atmospheric chemical processes, and the concentration of
$O_3$ was found to be related to the formation and growth of new particles at Station Nord during
summer based on case studies of NPF events in 2012 (**Fig. 9**).
*Event A,* **Fig. 6**: Event A is in fact a "double" event, with the first event occurring over June 15 - 16
followed by another event starting on June 17 with traceable growth until June 20.
During June 15, the $O_3$ level (black line) increased considerably to ~45 ppbv, which was
significantly higher than the average summer (June - August, 2012) concentration of $O_3$ (~26 ppbv).
As the NPF event on June 15 started followed by particle growth up to ~25 nm, the $O_3$ level
dropped dramatically, then somewhat stabilized when the approximate mean particle size reaches
the lower Aitken mode. The next drop in $O_3$ concentration (from ~37 ppbv to ~27 ppbv) coincided
with the occurrence of the second NPF event observed around noon of June 17. As the new particles
grew beyond ~30 nm in diameter, the $O_3$ concentration seemed to stabilize again.
In the late hours of June 19, the $O_3$ concentration suddenly dropped by ~5 ppbv, coinciding with an
interruption of the event. By midday June 20, the $O_3$ concentration increased back to the pre-
interruption level, while that interrupted event also seemed to be brought back to the station. It was
unclear if this drop of $O_3$ concentration on June 19 was associated with any NPF, as nucleation
sized particles were also observed for a few hours during early hours on June 20. However, a full
justification of this observation was not possible due to the detection limit of the Mobility Particle
Size Spectrometer system (~10 nm) confining to only aged nucleation particles. Another
explanation could be that both $O_3$ and the nucleation event were transported to the station from a
common source, with the interruption probably indicating for instance a displacement of air mass.
It has been observed that $O_3$ depletion occurs only when filterable bromide fBr is present, which is
in agreement with the evidence that $O_3$ is removed by Br atoms (Skov et al., 2004; Goodsite et al.,
2004; Goodsite et al., 2013). NPF at coastal location has also been found related to iodine oxides
(O'Dowd et al., 2002; McFiggans et al., 2010; Mahajan et al., 2011; Saiz-Lopez and von Glasow,
2012). This study was however unable to investigate the possible impact of halogen chemistry, due
to a lack of relevant measurement data.
During *Event A* case study, the NO and $NO_x$ level remained mostly below 0.1 ppbv. This was
approximately the background level of $NO_x$ at Station Nord throughout the year. NO and $NO_x$



concentration did not seem to relate to $O_3$ concentration level, or observations of new particle
formation events.
***Event B, Fig. 6***: This *Event B* on August 2 showed that a lower level of $O_3$ concentration (~25
ppbv) could also be associated with a new particle formation event. During the event, the episode of
traceable particle growth lasted for approximately 12h, coinciding with a concurrent drop of the $O_3$
concentration. This event was also considerably less intensive in regards of particle number
concentrations compared to *Event A*. Until the end of the event, particles were mostly below 30 nm
in size.
***Event C, Fig. 6***: During this event on August 9 - 10, new particle formation was also observed
together with lower $O_3$ concentrations (~25 ppbv), which was similar to *Event B*. The anti-
correlation between growth of newly formed particles and $O_3$ concentration was also observed
during this event. However, such anti-correlation was visible until particles almost reached 40-50
nm in diameter, which was higher than that observed during *Event A* and *Event B*. The growth
seemed to be interrupted in the morning of August 10, allowing the concentration of the $O_3$ oxidant
to recover during that exact period back to values above 25 ppbv.
As demonstrated with the three events, the concentration level of $O_3$ seemed to display an anti-
correlation with early particle growth up to about 30 nm during *Event A* and *Event B* or about 40-50
nm in case of *Event C*. It is generally agreed that particle nucleation involves sulfuric acid ($H_2SO_4$)
via the oxidation of $SO_2$ by the hydroxyl (OH) radical (Kulmala et al., 2001), while particle growth
depends considerably on vapor uptake and condensation of low-volatile organic vapor products
produced by photo-oxidation of volatile organic compounds (VOCs) (Donahue et al., 2011;
Riipinen et al., 2011; Riipinen et al., 2012). Naturally, $O_3$ is a major atmospheric oxidant, which
also undergoes photolysis to form the OH radical oxidant. These oxidants oxidize VOCs to form a
variety of low-volatile products. A reduction of $O_3$ could thus be an indirect indicator of increased
availability and thus uptake of low-volatile compounds, contributing to particle growth. Meanwhile,
it should also be noted that the role of halogen chemistry contributing to new particle formation is
unknown, due to a lack of relevant data as discussed above.
The source of VOCs at VRS, Station Nord is unclear. There might be some biogenic emissions of
VOCs at the station during summer, expected due to retreated snow and ice cover, exposed bare
ground and thus possibly increased biogenic activity. However, since this area is arid, this is
expected to be extremely limited. Meanwhile, the presence of VOC oxidation products such as



organic acids and organosulfates at the station has been reported by Hansen et al. (2014), though at
very low concentrations. The low mass loading of organic materials (Nguyen et al., 2014) and total
suspended particles (Nguyen et al., 2013) observed at the station during summer would inhibit
removal of small particles by coagulation, thus allowing particle growth and prolonged NPF events.
As $O_3$ only seemed to inversely correlate with particle growth up to aged nucleation or lower-
Aitken size, poor correlations were obtained between $O_3$ concentration and particle number
concentrations. Although the summer months in 2012 were event-active, the Pearson correlation
coefficients between $O_3$ concentrations and particle number concentrations during June, July and
August were 0.37, 0.26 and -0.16, respectively. Meanwhile, it was found that $O_3$ correlated
positively with the observed particle volume concentrations during June (0.80), July (0.57), August
(0.38) and September (0.50), which probably indicated that oxidation by $O_3$ was no longer
important as particles reached larger size. At the same time, the possibility of the $O_3$ oxidant and/or
the new particle formation events being transported to the site in the same or different air masses
cannot be eliminated and will be examined further using HYSPLIT analysis.
**NO$_x$**
As mentioned above, sparks of particle formation, which did not grow further, were considered as
local pollution events, which related to NO$_x$ emitted by the car engine during service of the station.
There was probably some additional contribution from emissions from the military base, which is
located at a distance of about 2.5 km from the measurement site. An example of such interference is
illustrated during the early hours of August 2 (*Event B*, **Fig. 6**), during which a higher NO$_x$
concentration of ~0.15 ppbv was detected together with a short episode of new particle formation
without further growth. Such interference could also be observed around midday of the same event
day (*Event B*, **Fig. 6**). In contrast, it must be noted that NO$_x$ concentrations in the range ~0.1-0.2
ppbv were mostly not associated with any noticeable observations of new particle formation.
During the late winter - spring months (March - May), episodes of depletion or complete removal of
the surface layer $O_3$ and mercury in the Arctic occur due to reaction with atmospheric bromine
released from sea-ice and surface snow (Barrie et al., 1988; Bottenheim et al., 1990; Pratt et al.,
2013; Abbatt, 2013; Abbatt et al., 2012; Skov et al., 2004). The concentration of $O_3$ then is so low
that it can no longer oxidize NO and $NO_2$. Local NO$_x$ emissions thus seemed to relate to the intense
burst of small particles which lasted for hours. Removal of these episodes resulted in several
noticeable gaps in the data set, especially in March and May 2012 (**Fig. 3**).



The summer period June - August was associated with a lower level of background $NO_x$ ($NO_x$ ~0.1
ppbv) compared to the rest of the year ($NO_x$ ~0.2 ppbv). $NO_x$ emissions into the Arctic atmosphere
other than the direct local anthropogenic emissions could originate from the thermal decomposition
of PAN, which is the major atmospheric $NO_x$ reservoir species (Singh et al., 1995). This process is
nevertheless limited by low temperature during winter and spring and low PAN levels during
summer (Beine and Krognes, 2000). $NO_x$ also contributes via photochemistry to the local formation
of tropospheric $O_3$ and thus enhances $O_3$ levels during summer (Walker et al., 2012; Beine and
Krognes, 2000) at the expense of $NO_x$ concentrations. However, a direct relation between $O_3$ and
$NO_x$ during summertime was not observed (**Fig. 6**).
**3.2.3 Analysis of air mass back trajectories**
As mentioned above, the Mobility Particle Size Spectrometer system employed at VRS, Station
Nord is limited to particles larger than 10 nm in size, capturing only aged nucleation particles. It is
thus uncertain whether the formation of the freshly nucleated particles actually occurred at the site,
or if they were transported from elsewhere or produced aloft.
Air mass back trajectories were analyzed in order to investigate possible source regions for the
observed events. The trajectories were calculated using HYSPLIT (Draxier and Hess, 1998). The
model runs were based on meteorological data obtained from the Global Data Assimilation System
(GDAS), which is maintained by the US National Centers for Environmental Prediction (NCEP).
Air mass back-trajectories were calculated 24h to 48h backwards for air masses arriving at the
station at 50 m and 500 m above sea level on the event days, which were discussed earlier in **Fig. 6**.
The trajectories were presented in **Fig. 7**, with the names of the events kept consistent with those in
**Fig. 6**. Only the first two days (June 15 - 16) and the last two days (June 19 - 20) of *Event A* was
shown in **Fig. 7**. Calculations of air mass back trajectories were performed for these two day
periods, in order to minimize the uncertainties associated with calculating trajectories many days
backwards.
Descending of air parcels from above the boundary layer was commonly observed on many event
days, such as during *Event A* (June 15 - 16, 2012) and *Event C* (August 2, 2012) (**Fig. 7**). Strong
vertical mixing could relate to an interruption of an event. For example, an episode of vertical
mixing between the lower (red) and upper air parcels (blue) occurred around mid-day of June 19,
2012 and lasted until the early morning hours of the following day (~15 hours in total) (**Fig. 7**). This
could probably relate to the interrupted phase of particle growth and $O_3$ concentration earlier



observed (~18 hours in total) (*Event A,* **Fig. 6**). The event interruption was also observed a few
hours later, which was probably due to the travelled distance of the air mass between the vertical
displacing point above the boundary layer and that reaching the station at the ground level.
Nevertheless, as *Event A* resumed after the interruption on June 20, 2012, the observed lower
Aitken mode band seemed to continue the growth before the interruption (**Fig. 6**). Such observation
probably indicated that the air parcels providing the source to the new particle formation events
(and possibility also $O_3$) could be displaced from and then brought back to the station.
Subsequently, this could indicate that the entire event was "transported" from aloft down to the
ground level. Similarly, during *Event B* (August 2, 2012), vertical mixing between the upper air
parcels (blue) and lower air parcels (red) occurred around noon time and lasted for ~12 hours (**Fig.**
**7**). This seemed to relate to the NPF event occurring around the same time with roughly the same
length (~12 hours) (*Event B*, **Fig. 6**).
In fact, it was previously indicated that new particles could be formed aloft and subsequently
transported to the ground level due to vertical mixing, leading to new particle formation events
observed around noon time (Mäkelä et al., 2000; Crippa et al., 2012; Pryor et al., 2010). In another
study by Wiedensohler et al. (1996), it was also suggested that the observed occurrence of particles
smaller than 20 nm in diameter in the marine boundary layer over the Arctic pack ice could
originate from higher altitudes. Assuming that the new particle formation events were transferred
from aloft, it is possible that the vertical mixing with the upper air parcels could either interrupt an
event or lead to observation of a new event at the site. This would depend on whether the displaced
air parcels or the displacing air parcels are event-active, or having the favorable conditions for the
formation and growth of new particles, such as the presence of precursor gases. In contrast, an
observed interruption during a new particle formation event such as during the early hours of
August 10, 2012 (*Event C*, **Fig. 6**) was not always related to displacing air parcels. The interruption
could instead relate to a change in the horizontal direction of the air parcels arriving at the station
occurring around midnight of August 9, 2012 (**Fig. 7**).
Air mass back trajectories were also calculated three-days backwards, at one hour after the starting
time of each identified event using HYSPLIT, whereas for the other days, trajectories arriving at
12:00 p.m. local time were used. The region around Station Nord was split into one degree
latitudinal and six degree longitudinal grid boxes. Every time a trajectory passed one grid box, a
count was registered for that grid box. The probability of registering an event, when the air mass
originated from a specific grid box was obtained by dividing the total counts during event days by





the sum of total counts during event days, undefined and non-event days. The probability results are
shown in **Fig. 8**.
As apparent from the figure, the probability of observing an event at the station is low when the air
masses arrive from the southwesterly direction over Greenland. Other directions of air mass origin
however showed relatively similar probability of registering an event. A slightly higher probability
range was observed for southeasterly air masses that passed over region, where open waters and
melting ponds on ice are more likely to occur. As particles typically grow very slowly at Villum
Research Station, the time gap from particle nucleation occurring around 1.5 nm in diameter until
the point when they are observed at the site (~10 nm in diameter) could range from hours to days.
The corresponding probability for observing nucleation mode particles (~10 nm in diameter) at the
site should therefore serve as an indication of probable air mass origin of the grown nucleation
mode instead of freshly nucleated particles.
**3.2.4. Analysis of wind pattern during NPF events**
The wind pattern was also investigated on specific event days in 2011 and 2012 (figure not shown).
However, they were found very similar to the general wind patterns of the corresponding month or
period. Therefore, it is unlikely that any change in local wind direction during the specific event
days could have an impact on the occurrence of new particle formation events observed at the site.
This indicates the possibility of other factors, which may have changed during the event days
affecting new particle formation such as precursors. In fact, Quinn et al. (2002) indicated that the
abundant dimethyl sulfide (DMS) could affect particle production during summer, as evidenced by
a strong correlation between particle number concentrations and methanesulfonate ($MSA^-$)
concentrations (resulting from the oxidation of DMS). Similar observations were reported by
Leaitch et al. (2013). Other examples of factors influencing NPF are atmospheric oxidation capacity
and transport of air masses.
**3.2.5. Event statistics**
In general, the event days accounted for 15 - 38 % of the classified days during June - September,
with the highest percentages of event days observed in August (38 %) and July (33 %) (**Table 3**).
The period from June to early September was also the period during which longer events up to
several days were observed and most class I events were identified (**Table 3**).
The observed frequencies of event days during these months at VRS, Station Nord were relatively
higher compared to reported values from sub-Arctic stations during the same months, such as



Värriö (20 - 25%) (Kyro et al., 2014), Pallas (10 - 20 %) (Asmi et al., 2011) or Abisko (< 20 %)
(Vaananen et al., 2013). In fact, the observed new particle formation events at these stations and
other Nordic stations seemed to show a spring maximum of event occurrence (Vehkamaki et al.,
2004; Dal Maso et al., 2007; Kristensson et al., 2008), as opposed to the summer maximum of
events observed at VRS, Station Nord. At the same time, NPF events were still observed at the sub-
Arctic stations Värriö , Pallas and Abisko during the darker months (November - February), though
the fraction of event occurrence was typically much lower compared to other seasons (Kyro et al.,
2014; Asmi et al., 2011; Vaananen et al., 2013). Notably, not a single event was observed at VRS,
Station Nord during the Arctic night in the absence of sunlight.
**4. Conclusion**
In this work, the seasonality of particle number size distributions, total particle number, volume and
mass concentrations was examined. A strong seasonal pattern was found, showing the abundance of
smaller particles during the sunlit period of the year, especially during summer and a persistent
accumulation mode during the darker months caused by long-range transport of particles to the
Arctic. Analysis of wind data showed a dominance of easterly winds during the summer months and
southwesterly winds during the darker months of the year.
NPF events were investigated based on case studies, showing clear events lasting from hours to
days. $O_3$ was found closely related to the NPF events observed at the station, especially in regards
of particle growth. Calculations of air mass back trajectories on the days with new particle
formation events using HYSPLIT indicated an aloft origin of air parcels arriving at the station on
many event days. The overlaps between the occurrence of vertical displacing air masses and
interruption of events observed at the measurement site further suggested that the event could be
transported to or displaced from the site together with the air masses. Air masses arriving from the
southwesterly direction over Greenland were least linked to NPF event, whereas air masses arriving
from southeasterly direction over Greenland sea was associated with slightly higher probabilities.
Meanwhile, the local wind direction did not seem to relate to NPF events observed at the station
**Acknowledgements**
This work was financially supported by the Danish Environmental Protection Agency with means
from the MIKA/DANCEA funds for Environmental Support to the Arctic Region, which is part of
the Danish contribution to "Arctic Monitoring and Assessment Program" (AMAP) and to the
Danish research project "Short lived Climate Forcers" (SLCF). The findings and conclusions



presented here do not necessarily reflect the views of the Agency. This work was also supported by
the Nordic Centre of Excellence Cryosphere-Atmosphere Interactions in a Changing Arctic Climate
(CRAICC). The Villum Foundation is acknowledged for funding the construction of Villum
Research Station, Station Nord. The authors are also grateful to the staff at Station Nord for their
excellent support.





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





**List of Figures**
**Fig. 1.** The high Arctic site Villum Research Station, Station Nord (81°36' N, 16°40'W, 30 m a.s.l.)
in northeast Greenland. The main measurement site is Flyger's hut, which is located about 2.5 km
southeast of the Danish military base.
**Fig. 2.** SMPS, $O_3$ and NOx data coverage at Station Nord from July 2010 - February 2013.
**Fig. 3.** Time series of particle number size distributions as dN/dlogDp $(cm^{-3})$ during 2012. The
original 5 min time resolution was used in the plots.
**Fig. 4**. Monthly median particle number size distribution at Station Nord during 2012. The
corresponding lognormal-fitting parameters are shown in **Table 2**.
**Fig. 5.** Windroses showing monthly wind direction and wind speed at Station Nord during 2012.
The concentric rings show the percentage of wind arriving from a particular direction.
**Fig. 6**. Demonstration of the impacts of $O_3$, NO and $NO_x$ on the summer new particle formation
events occurring on June 15-20 (Event A), Aug 2 (Event B) and Aug 9-10 (Event C) in 2012.
**Fig. 7.** Demonstration of air mass back trajectories calculated using HYSPLIT for arrival at 50 m
and 500 m at the station on selected days with new particle formation events.
**Fig. 8.** The probability of observing an event at Station Nord (bottom tip of the black triangle) as a
function of air mass origin.
**Fig. 9.** Monthly variation of total number of days with good data (left vertical axis) and frequency
percentages (%) of event days, non-event days and undefined days (right vertical axis) during the
study period (July 2010 - February 2013).





**List of Tables**
**Table 1.** Three modes were fitted to the average monthly data of 2012 using lognormal fitting. The
parameters shown for each mode include the modal number concentration (N, cm$^{-3}$), the modal
geometrical mean diameter ($D_g$, nm) and the modal geometrical standard deviation (GSD).**Table 2.**
Three modes were fitted to the average data for each month of 2012 using lognormal fitting. The
parameters shown for each mode include the modal number concentration (N, cm$^{-3}$), the modal
geometrical mean diameter ($D_g$, nm) and the modal geometrical standard deviation (GSD).
**Table 2.** Median and average particle number concentration (N), particle volume concentration (V)
and particle mass concentration (M) for the 12 months of 2012. M was calculated from V assuming
a density of 1.4 g cm$^{-3}$ and particle sphericity.
**Table 3.** Percentage of total new particle formation events (marked in blue) versus non-events and
undefined days during the period July 2010 to February 2013. The total events were further divided
into Class I and Class II events. A column of total days (by month) over the studied years was also
provided.



# 1  Figures

**Fig. 1.** The high Arctic site Villum Research Station, Station Nord (81°36' N, 16°40'W, 30 m a.s.l.) in northeast Greenland. The main measurement site is Flyger's hut, which is located about 2.5 km southeast of the Danish military base.

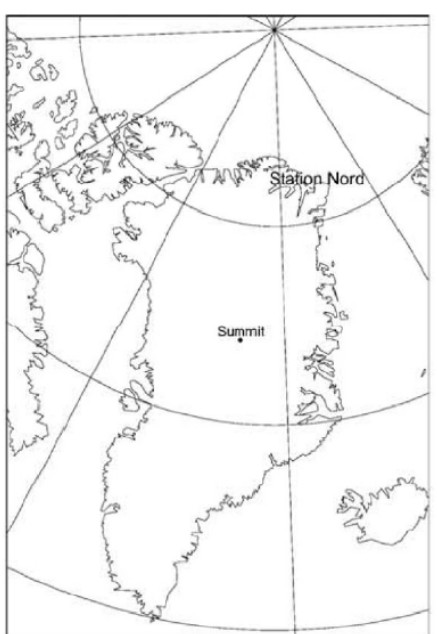
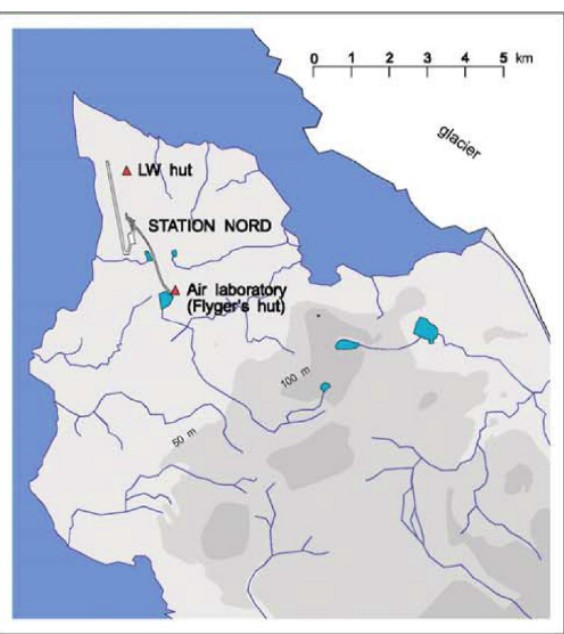



1    **Fig. 2.** SMPS, $O_3$ and $NO_x$ data coverage at Station Nord from July 2010 - February 2013.

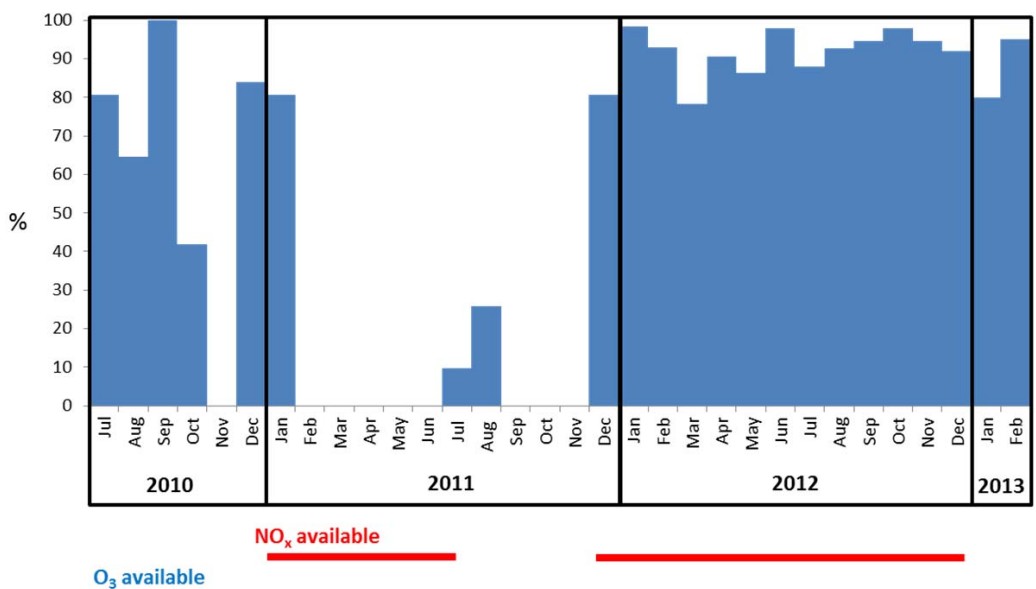



1 **Fig. 3.** Time series of particle number size distributions as dN/dlogDp (cm$^{-3}$) during 2012. The
2 original 5 min time resolution was used in the plots.





**Fig. 4**. Monthly median particle number size distribution at Station Nord during 2012. The corresponding lognormal-fitting parameters are shown in **Table 2**.





1    **Fig. 5.** Windroses showing monthly wind direction and wind speed at Station Nord during 2012.

2    The concentric rings show the percentage of wind arriving from a particular direction.



1 **Fig. 6.** Demonstration of the connection between $O_3$, NO and $NO_x$ and summertime new particle
2 formation events occurring on June 15-20 (Event A), Aug 2 (Event B) and Aug 9-10 (Event C) in
3 2012.

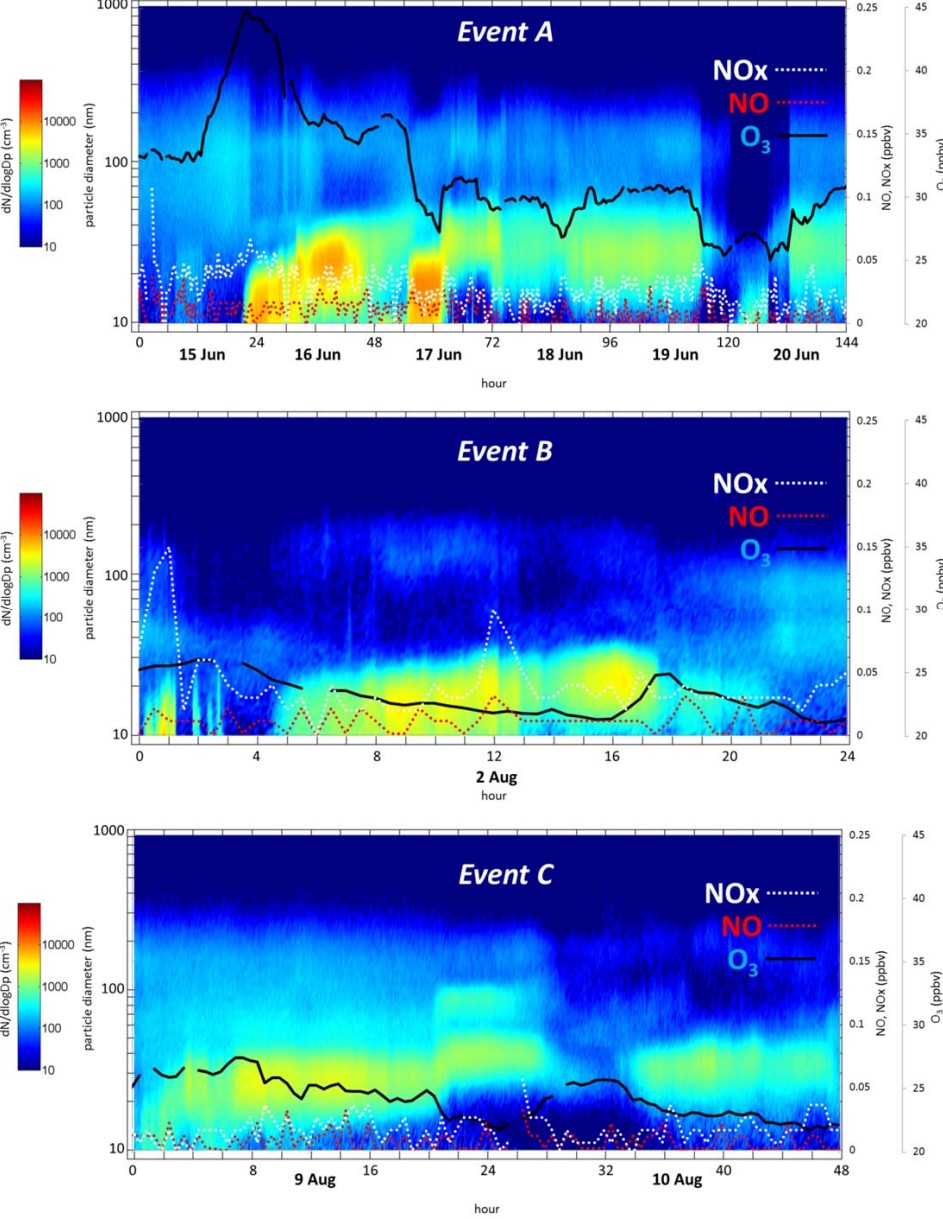





1   **Fig. 7.** Demonstration of air mass back trajectories calculated using HYSPLIT for arrival at 50 m
2   and 500 m at the station on selected days with new particle formation events.

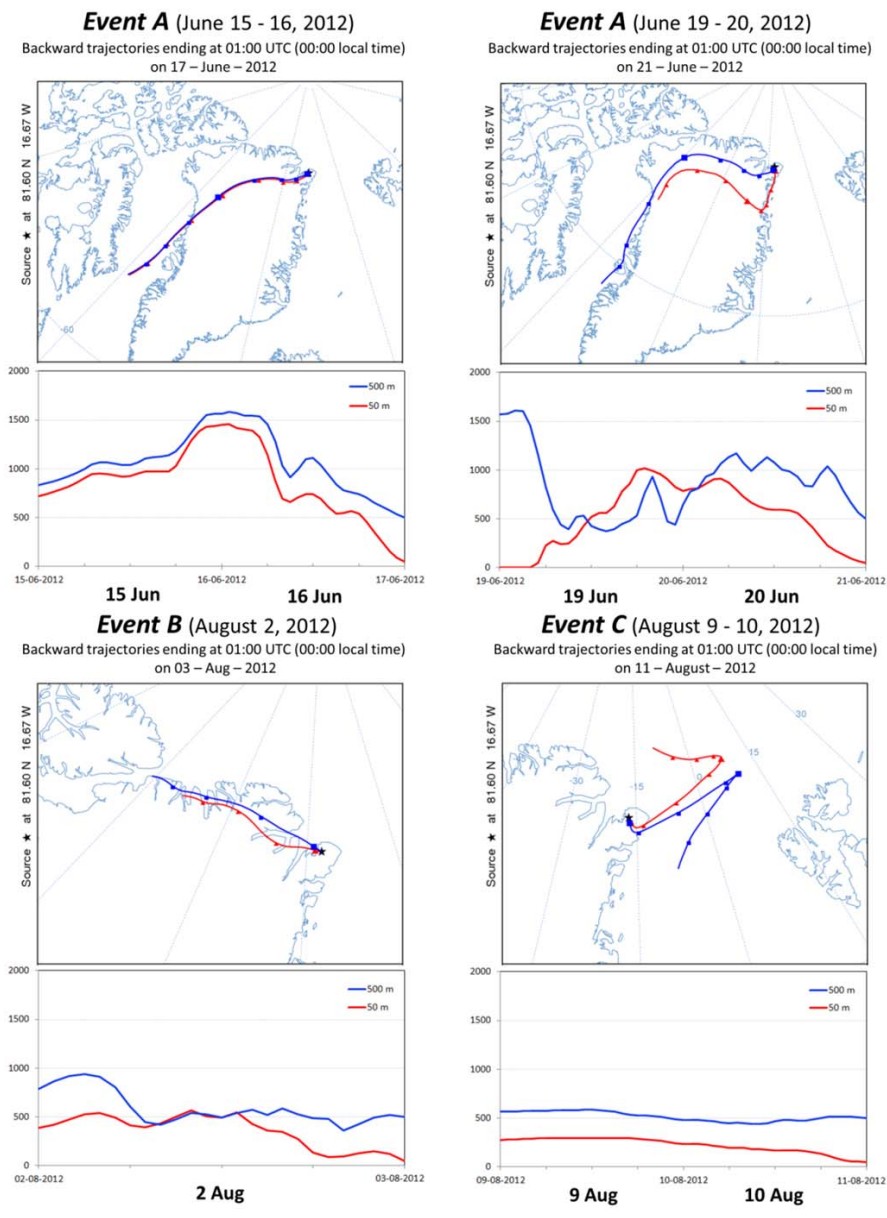



1   **Fig. 8.** The probability of observing an event at Station Nord (bottom tip of the black triangle) as a
2   function of air mass origin.

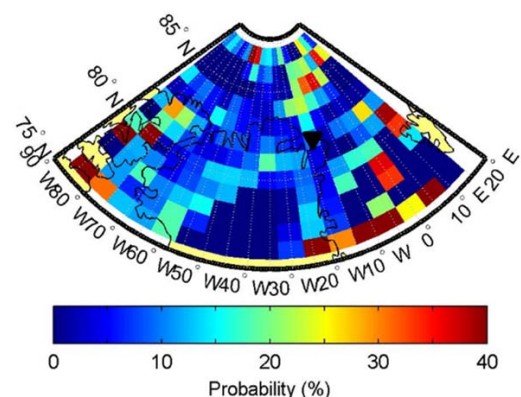





**Fig. 9**. Monthly variation of total number of days with good data (left vertical axis) and frequency
percentages (%) of event days, non-event days and undefined days (right vertical axis) during the
study period (July 2010 - February 2013).

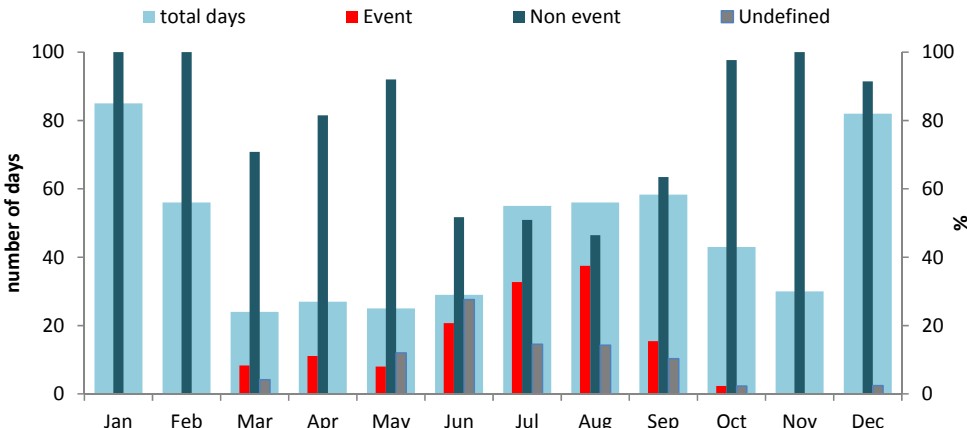




# 1   **Table**

2   **Table 1.** Three modes were fitted to the average monthly data of 2012 using lognormal fitting. The

3   parameters shown for each mode include the modal number concentration (N, cm$^{-3}$), the modal

4   geometrical mean diameter (D$_g$, nm) and the modal geometrical standard deviation (GSD).

| | $N_1$ (cm$^{-3}$) | $D_{g,1}$ (nm) | $GSD_1$ | $N_2$ (cm$^{-3}$) | $D_{g,2}$ (nm) | $GSD_2$ | $N_3$ (cm$^{-3}$) | $D_{g,3}$ (nm) | $GSD_3$ |
|---|---|---|---|---|---|---|---|---|---|
| January | 5 | 22 | 1.4 | 72 | 68 | 3.3 | 50 | 167 | 1.6 |
| February | 22 | 27 | 2.2 | 58 | 97 | 2.7 | 75 | 169 | 1.5 |
| March | 24 | 17 | 1.7 | 49 | 84 | 2.8 | 93 | 179 | 1.7 |
| April | 45 | 24 | 2.4 | 38 | 48 | 1.6 | 172 | 167 | 1.6 |
| May | 17 | 18 | 1.2 | 134 | 43 | 2.5 | 125 | 173 | 1.5 |
| June | 252 | 17 | 1.9 | 22 | 31 | 1.4 | 45 | 113 | 1.5 |
| July | 196 | 21 | 2.6 | 24 | 45 | 1.3 | 50 | 119 | 1.6 |
| August | 287 | 16 | 2.3 | 51 | 30 | 1.5 | 49 | 114 | 1.8 |
| September | 90 | 11 | 1.5 | 25 | 29 | 1.4 | 57 | 107 | 1.8 |
| October | 25 | 9 | 1.3 | 60 | 41 | 3.3 | 24 | 139 | 1.5 |
| November | 12 | 16 | 1.7 | 45 | 62 | 2.6 | 51 | 173 | 1.5 |
| December | 31 | 22 | 2.4 | 48 | 100 | 2.5 | 35 | 170 | 1.5 |



1  **Table 2.** Median and average particle number concentration (N), particle volume concentration (V)

2  and particle mass concentration (M) for the 12 months of 2012. M was calculated from V assuming

3  a density of 1.4 g cm$^{-3}$ and particle sphericity.

| | Median N (cm$^{-3}$) | Average N (cm$^{-3}$) | Median V (µm$^3$ cm$^{-3}$) | Average V (µm$^3$ cm$^{-3}$) | Median M (µg m$^{-3}$) | Average M (µg m$^{-3}$) |
|---|---|---|---|---|---|---|
| January | 104 | 121 | 0.44 | 0.69 | 0.61 | 0.96 |
| February | 123 | 149 | 0.69 | 0.82 | 0.97 | 1.15 |
| March | 170 | 174 | 1.10 | 1.13 | 1.54 | 1.58 |
| April | 231 | 253 | 0.88 | 0.93 | 1.24 | 1.30 |
| May | 221 | 268 | 0.78 | 0.78 | 1.09 | 1.09 |
| June | 137 | 277 | 0.14 | 0.15 | 0.20 | 0.21 |
| July | 229 | 237 | 0.17 | 0.20 | 0.23 | 0.29 |
| August | 227 | 313 | 0.19 | 0.21 | 0.27 | 0.29 |
| September | 124 | 137 | 0.18 | 0.18 | 0.25 | 0.25 |
| October | 71 | 87 | 0.17 | 0.25 | 0.24 | 0.35 |
| November | 96 | 100 | 0.40 | 0.42 | 0.55 | 0.59 |
| December | 85 | 107 | 0.30 | 0.57 | 0.42 | 0.80 |



**Table 3.** Percentage of total new particle formation events (marked in blue) versus non-events and
undefined days during the period July 2010 to February 2013. The total events were further divided
into Class I and Class II events. A column of total days (by month) over the studied years was also
provided.

|  | Total days | Class I (%) | Class II (%) | Total events (%) | Non-events (%) | Undefined (%) |
|---|---|---|---|---|---|---|
| January | 85 | 0 | 0 | 0 | 100 | 0 |
| February | 56 | 0 | 0 | 0 | 100 | 0 |
| March | 24 | 0 | 8 | 8 | 71 | 4 |
| April | 27 | 0 | 11 | 11 | 81 | 0 |
| May | 25 | 0 | 8 | 8 | 92 | 12 |
| June | 29 | 7 | 14 | 21 | 52 | 28 |
| July | 55 | 9 | 24 | 33 | 51 | 15 |
| August | 56 | 9 | 29 | 38 | 46 | 14 |
| September | 58 | 5 | 10 | 15 | 63 | 10 |
| October | 43 | 0 | 2 | 2 | 98 | 2 |
| November | 30 | 0 | 0 | 0 | 100 | 0 |
| December | 82 | 0 | 0 | 0 | 91 | 2 |
| Total | 570 | 3 | 9 | 11 | 79 | 7 |

