# Peer review of "Seasonal variation of atmospheric particle number"

_Atmospheric Chemistry and Physics, 2016_

## Referee Comment (RC1) · Anonymous Referee #1 · 19 May 2016

Overview:

This study presents aerosol size distribution data collected at VRS Nord. The main focus is on new particle formation. The subject is well within the scope of ACP, and especially new particle formation remain a challenge to resolve in this region.

Although the topic is relevant indeed, in its current form the MS suffers from several shortcomings.

General comments:

[Figure]

The analysis performed is in many aspects superficial, and does not leave much support to the conclusions presented. Thus, the MS need substantial improvements prior final publication. My main concerns are listed below, followed by more detailed comments.

I suggest providing a more detailed description of the whole data set. The authors has a large number of events to derive statistics from, but still only treats 3 specific events in detail. It is ok ta have case studies, but I do miss the general picture regarding timing of events, role of ozone and trajectory analysis. What is the role of mass and surface during onset of events?

The analysis of the role of ozone is superficial and is apparently based on three case studies. In the MS, it seems that the role of ozone is discussed in an unjustified way. More detailed analysis is required to create support statements regarding the role of o3.

The interpretation of the trajectory analysis regarding the three cases A-C is conceptually wrong. Suggestions for improvements of analysis is given under specific comments below.

Currently, two of the main conclusions provided by the authors lacks adequate support from the analysis performed. In its current form, it is not recommended to proceed with publication of the MS.

However, as information regarding new particle formation is scarce in the Arctic, as also acknowledged by the authors, I do not want to reject the MS completely. However, major revisions is needed prior publication, taking into account the comments given below.

Recommendation: Major revision

Specific comments:

Page 2, line 3: Not true. Accumulation mode is apparently present, but in low concentrations.

Page 4, line 15: Unsure if "while" is proper word here.

Page 4, line 20: "Details"→"detail"

Page 5, Instrumentation: I assume that correction for diffusion losses and CPC efficiency is taken into account during the inversion, but suggest that it is explicitly stated as the paper is dealing with new particle formation.

Page 5, Instrumentation: Please provide specification of the inlet used.

Page 6,line 9-10: Sentence awkward, suggest rewording.

Page 7, line 19: What data is actually used in this study, only 2012 or the period 2010-2012.cf page 6, line 13.

Page 7, Figure 4: I suggest adding some range to the data in the figure, i.e. percentile ranges in some suitable range, i.e. 25th-75th percentile to give a better view of variability.

Figure 4: Caption: Should say "table 1", not "table 2" if text is referring to lognormal-parameters. Please also specify how the fitting was done

Page 8, line 15: Why does the figure "especially" describe the accumulation mode? Suggest rewording

Table 2: Is there a special reason for selecting a density of 1.4g/cm3

Page 8, line 26-29: Suggest rewording. As it reads now, it seems that the nucleation mode concentration is attributed to high accumulation mode concentration.

Page 10, line 27: Suggest replace "high" with "strong" or similar.

Page 10, line 29: At what altitude above ground is the anemometer placed. How does topography around the site of observation affect the quality of observations? Why weren't trajectories used to classify the transport?

Page 10, line 31: What do the authors mean by a "broader region"?

Page 11, section 3.2.1: I suggest that the authors provide a table or figure showing timing of onset of nucleation, a simple frequency diagram would suffice. This will show if there is any diurnal preferences regarding onset of NPF.

Page 12, line 4-5: It is unclear what the authors mean here. Figure 9 does not show any details about ozone, why it is difficult to follow the authors reasoning. I strongly suggest that the authors provide a more detailed evaluation of the role of O3 in NPF or tracer thereof.

Table 3: The numbers do not add up: I assume that data is divided in three main classes: events, non-events and unclassified. Together these should add up to 100%. However, reading the table, March has 83% total, April 92% total, May 112% total. Thus, it needs to be clarified to what the percentage is referring to if not to total amount of observations!?

Page 12, line 15-16: It seems that the station is within a cloud/fog. Is the inlet whole air or does it have a well defined cut off?

Page 12, line 24-29: Suggest deleting this paragraph as it has nothing to do with NPF.

Page 13, line 13-15: What does the author mean by "allowing the ozone to recover"? The reasoning seem to assume that the growth control ozone and not vice versa... To me, it seems like a change in air-mass.

Page 13, line 16-20. This is a quite shaky statement. The variation in ozone and relation formation/growth is not unambiguous. Further, the statement is based on three cases. With less than the authors back the discussion with more statistics, to me the role of ozone seems speculative to say the least. Thus, I strongly suggest that the relation to ozone is studied more in detail. In its current form however, the limited amount of data presented doesn't support anything.

Page 14, line 2-4: The author discuss the role of mass (or rather surface) in regulating

amount of small particles via coagulation. Of higher concern would be the role of the condensation sink (CS) of both nucleating and condensing species.

Page 14, first section: Is the correlation derived for the integral number or for number concentration over some representative size interval? If the latter is not true, I suggest an integration over nuclei mode size range, e.g. 10-30nm, prior attempting correlation tests.

Page 14, line 28-31: I do not follow completely. As I read the text, low ozone hinders NO-NO2 transition. So far so good. However, then the author state: "Local NOx emissions thus seemed to relate to...". What do the authors mean?? What is the connection?

Page 15, line 8: "...at the expense of NOx concentration..." I was under the impression that the role of NOx in ozone formation is the photolysis of NO2 to NO+O. Thus, ozone formation does not reduce total levels of NOx.

Page 15, line 26: On how big fraction did this occur? How is the boundary layer defined?

Whole discussion using trajectories: The approach using the trajectories is conceptually wrong. All four examples is calculated for trajectory arrival around 01:00 at the end of the event. How does that at all relate to the onset/evolution of the event. In order to be able to discuss anything about the role of the air-mass to some aerosol property, arrival time of trajectory and observation of the parameter must coincide in time. The problem in reasoning is exemplified by e.g. Page 15, lines 28-onwards where the authors discuss mixing along the trajectories. It has to be made clear, that the mixing taking place during 19-june in a trajectory arriving 21-june, 01:00 absolutely has nothing to do with the air arriving to the site 19-june. At least, the author has not shown that such a connection exist.

In order to do the analysis properly, I would recommend the authors to look at the

whole time series during the event, calculate one trajectory every hour for the duration of the event. Only then aerosol properties during the event will be linkable to air-mass transport.

As currently used, the conclusions from the trajectory analysis down to page 16, line 26, are completely redundant and should be removed or revised accordingly.

Page 18, conclusions: The conclusion that ozone is closely related to NPF is not supported by the analysis presented. Three presented cases, and a typically poor correlation between ozone and particle integral number, is clearly not enough. The same goes for the conclusions from the trajectory analysis regarding subsidence. The approach by which the authors derive the conclusion is wrong. Thus, the only verifiable outcome of the analysis is seasonality of NPF events.

---

## Referee Comment (RC2) · Anonymous Referee #2 · 23 May 2016

The manuscript presents an analysis of aerosol characteristics at the high-Arctic site Station Nord in Greenland, based on continuous measurements during 2010–2013 (concentrating on year 2012). The focus of the manuscript is in analysis of new particle formation (NPF) events. Ambient conditions favoring NPF at the site are reported based on case-studies, and NPF events are also analyzed with respect to source areas based on airmass back-trajectories. The dataset presented in the manuscript is interesting, higlighting the importance of atmospheric NPF to the aerosol number even in the remote Arctic regions. This work is within the scope of Atmospheric Chemistry and Physics, and could be considered for publication after the comments below have

been taken into account.

My major concern is the analysis of the airmasses in relation to the three NPF events presented in the paper. The arrival times of the airmass back-trajectories shown in Figure 7 do not seem to coincide with the NPF events presented in Figure 6. The first airmass shown arrives in-between of the double event A and all the other airmasses of Fig. 7 on the midnight following the events A, B and C. I don't see how any conclusions on aerosol and trace gases measured at Station Nord during these three NPF events could be drawn based on the airmasses shown in Fig 7. Therefore, the analysis of the whole of Section 3.2.3 should be redone by analysing airmasses arriving to the station at the start of the NPF event or at some other relevant time during the event.

Other general comments:

1) Page 2, lines 3–4: "only nucleation and Aitken-mode particles were observed during the summer months". Based on Figures 3–4 and Table 1, this does not seem to be the case. There are clearly particles larger than 100 nm present during all the months, although the concentrations of accumulation mode particles are lower during the summer.

2) Also Asmi et al. (2016) reported on NPF observations at the Arctic measurement station Tiksi in northern Siberia. This could be added to the discussion on NPF observations in the Arctic (third paragraph of the Introduction section).

3) Page 4, lines 32–33: Is the local pollution source taken into account in the data-analysis (for example by excluding data when the local wind direction is from the sector towards the pollution source)?

4) Page 7, line 19: "during the time period from July 2010 to February 2013". I suppose this should be "during 2012", as was stated at the end of Section 2.2.2. Also, Figures 3–5 refer to the year 2012.

5) Page 8, line 27: This sentence is little unclear, consider revising to e.g. "Since

nucleation mode particles were almost absent in April and relatively minor in May, the high median or average N values observed during these months were attributed to ..”

6) Could the analysis of Section 3.2.2 on the role of O3 and NOx in NPF made more general by including data during all the NPF events, instead of just using 3 case-studies? Also, comparison of the O3 and NOx between NPF and non-NPF days could provide useful information. Such analysis should probably be done seasonally in order to exclude the strong difference in NPF occurince between summer and winter.

7) Page 14, lines 29–31: What were the criteria used in the removal of the local pollution episodes? An exceedance of certain NOx level? Why weren't the episodes during August 2nd (Fig. 6) removed from the dataset, if they were identified as local pollution as discussed on lines 19–24 of page 14?

8) Page 15, lines 26–27: Are you certain that the airmasses descend from above the boundary layer in these two cases? At least for Event C the airmass arriving at 50 m height stays constantly below 250 m, which seems quite low to be above the boundary layer.

9) Page 15, lines 28–30: I don't fully understand how can the vertical mixing of the airmasses be inferred from Fig. 7 for the case of mid-day of June 19. According to the map, the two airparcels do not follow the same horizontal path, so even though they are at the same altitude at the same time on mid-day of June 19, they are not at the same location horizontally and therefore not interacting with each other.

10) Page 17, lines 1–2: Is the map of Figure 8 constructed using all the trajectories arriving at Station Nord during the year 2012? Is the number of trajectories big enough for drawing conclusions on the source areas of airmasses favouring NPF?

11) Page 17, lines 30–31: Asmi et al. (2016) reported similar NPF day frequency, 30–40%, during summer in Tiksi, Russia.

12) In the conclusions, the statement on the close relationship of ozone to the particle

growth (lines 18–19) seems hard to justify on the basis of the presented material, which is currently 3 case studies of NPF. What were the exact growth rates of 10–25 nm particles during these events? Could this analysis be made more thorough by including all the NPF days and showing the relationship between O3 concentration and particle growth rates (see also my comment 6)?

Technical comments:

Page 1, line 23: "focus" should be "focuses"

Page 6, line 16: section number "2.2.2" should be "2.2.3"

Page 12, line 5: "Fig. 9" should be "Fig. 6"

Page 17, line 30–31: ".. relatively higher compared to .." should be "relatively high compared to .."

References:

Asmi, E., Kondratyev, V., Brus, D., Laurila, T., Lihavainen, H., Backman, J., Vakkari, V., Aurela, M., Hatakka, J., Viisanen, Y., Uttal, T., Ivakhov, V., and Makshtas, A.: Aerosol size distribution seasonal characteristics measured in Tiksi, Russian Arctic. Atmos. Chem. Phys. 16, 1271–1287, 2016

---

## Author Comment (AC1) · 15 Jun 2016

Overview:

This study presents aerosol size distribution data collected at VRS Nord. The main focus is on new particle formation. The subject is well within the scope of ACP, and especially new particle formation remain a challenge to resolve in this region. Although the topic is relevant indeed, in its current form the MS suffers from several shortcomings.

General comments:

The analysis performed is in many aspects superficial, and does not leave much support to the conclusions presented. Thus, the MS need substantial improvements prior final publication. My main concerns are listed below, followed by more detailed comments.

I suggest providing a more detailed description of the whole data set. The authors has a large number of events to derive statistics from, but still only treats 3 specific events in detail. It is ok ta have case studies, but I do miss the general picture regarding timing of events, role of ozone and trajectory analysis. What is the role of mass and surface during onset of events?

The analysis of the role of ozone is superficial and is apparently based on three case studies. In the MS, it seems that the role of ozone is discussed in an unjustified way. More detailed analysis is required to create support statements regarding the role of O3.

The interpretation of the trajectory analysis regarding the three cases A-C is conceptually wrong. Suggestions for improvements of analysis is given under specific comments below.

Currently, two of the main conclusions provided by the authors lacks adequate support from the analysis performed. In its current form, it is not recommended to proceed with publication of the MS.

However, as information regarding new particle formation is scarce in the Arctic, as also acknowledged by the authors, I do not want to reject the MS completely. However, major revisions is needed prior publication, taking into account the comments given below.

Authors' response: Dear Reviewer,

Thank you for your general comments and concerns related to our manuscript. We have revised and improved the manuscript following the points that you have raised.

Specifically,

- We have provided an extra Supplementary Figure 1 related to the onset of events
- We have analyzed the role of ozone (by correlating O3 and particle number concentration of nucleation mode (10-30 nm) particles) for all events (35 cases) observed in 2012. Thereafter we found 46% of total events showing a weak to moderate anti-correlation (with Pearson correlation coefficient of $-0.71$), whereas the 54% did not show any correlation, and no positive correlation was found. Please see details of how we performed this analysis in text and also in our response to Comment 22.
- We have re-performed the trajectory analysis to hourly resolution, and to correctly match the event timing. The entire trajectory section has been re-written
- We have performed calculation of particle mass and surface during onset of events. While no difference was found, we have reported this in text accordingly.

Recommendation: Major revision

Specific comments:

**Comment 1**

Page 2, line 3: Not true. Accumulation mode is apparently present, but in low concentrations.

Authors' response: Thank you for noting this. We have revised the sentence accordingly.

**Comment 2**

Page 4, line 15: Unsure if "while" is proper word here.

Authors' response: "while" does not sound good here as a matter of fact. Thank you for reading our manuscript so carefully. We have revised this sentence accordingly.

**Comment 3**

Page 4, line 20: "Details"!"detail"

Authors' response: This has been revised accordingly.

**Comment 4**

Page 5, Instrumentation: I assume that correction for diffusion losses and CPC efficiency is taken into account during the inversion, but suggest that it is explicitly stated as the paper is dealing with new particle formation.

Authors' response: Thank you for being specific. We have added the following sentence to the section. "Specific DMA transfer function was used for inverting the data, while CPC efficiency and diffusion losses were corrected for during the inversion"

**Comment 5**

Page 5, Instrumentation: Please provide specification of the inlet used.

Authors' response: We have added the paragraph below to the end of the "Instrumentation" section accordingly.

Sampling was provided from a conductive flow tube. An air blower was used to suck a main air flow (much higher than the sample flow) into the main sampling inlet, and the air sampling was probed from this main air flow using a ¼ inch tubing directed into the main air flow. The main sampling inlet was not heated; however no icing issue was observed for the inlet. The main sampling inlet did not have any size cut-off. Sampling was performed at standard conditions of about 20 °C.

**Comment 6**

Page 6, line 9-10: Sentence awkward, suggest rewording.

Authors' response: This was indeed a fragmented sentence. It has been revised accordingly.

**Comment 7**

Page 7, line 19: What data is actually used in this study, only 2012 or the period 2010-2012.cf page 6, line 13.

Authors' response: We apologize that this was probably not clearly stated enough. Data from the other years were also used in Figure 8, 9 and Table 3 (together with data from 2012). It should already say on the figure/table captions, but we have added this sentence on page 6 about our use of data from the other years to make it explicit:

"Data from the other years were used to support the analysis of event statistics. Details of the data period used are provided in the caption of the relevant tables or figures."

**Comment 8**

Page 7, Figure 4: I suggest adding some range to the data in the figure, i.e. percentile ranges in some suitable range, i.e. 25th-75th percentile to give a better view of variability.

Authors' response: Thank you for your suggestion. We have added the $25^{th}$-$75^{th}$ percentile range to this Figure 4 accordingly.

**Comment 9**

Figure 4: Caption: Should say "table 1", not "table 2" if text is referring to lognormal parameters. Please also specify how the fitting was done.

Authors' response: Thank you for noting this. We have now revised caption of Figure 4 to correctly say "Table 1". We have also added this sentence in Table 1 caption to further explain how the fittings were done.

"A fitted sum of three lognormal distributions was calculated for the entire particle size range (averaged monthly particle number size distributions) and the difference of the sum of the squares of each number concentration at the specific sizes between the real and the fitted data was minimized using the Excel solver add-in."

**Comment 10**

Page 8, line 15: Why does the figure "especially" describe the accumulation mode? Suggest rewording

Authors' response: Thank you. This has been reworded to "showing". We agree that it sounds a lot better this way.

**Comment 11**

Table 2: Is there a special reason for selecting a density of 1.4g/cm3

Authors' response: Actually we were wondering ourselves about this. Arctic aerosols are quite aged and contain a lot of sulfates (if long-range transported) and thus would have a density higher than 1.4 g/cm3, especially during the Arctic haze period occurring yearly in late winter and spring. Tunved (2013) for instance used a density of 1 g/cm3, which in our opinion could be too low for those months.

On the other hand, it would be challenging to find a density that would represent the situation for the whole year. We have therefore selected a "moderate" density and clearly state our assumed value, so it could be compared with in other studies.

**Comment 12**

Page 8, line 26-29: Suggest rewording. As it reads now, it seems that the nucleation mode concentration is attributed to high accumulation mode concentration.

Authors' response: This was indeed a bad sentence. We have removed it completely.

**Comment 13**

Page 10, line 27: Suggest replace "high" with "strong" or similar.

Authors' response: Thank you for your suggestion. We have changed it to "strong" instead.

**Comment 14**

Page 10, line 29: At what altitude above ground is the anemometer placed. Lotte please help. How does topography around the site of observation affect the quality of observations? Why weren't trajectories used to classify the transport?

Authors' response: We have added a few extra sentences below to Section 2.2.3 to provide extra background on our wind measurements.

"The sonic is placed on a horizontal boom at the top of a 9 meter mast. The mast is situated about 36 m east-southeast from the measurement hut at ca. 62 meter asl. This means that the fetch limited wind direction is 300 degree where the hut (2.8 m) is an obstacle. The area is flat for 10-20 km in all wind directions."

On a local scale, our wind measurements should be reliable. However, as air masses arriving at the station tend to originate from the considerably longer distance, sometimes from another direction as indicated by HYSPLIT, we figured that we should analyze both wind and HYSPLIT back trajectories to get different perspectives on where the particles might come from.

**Comment 15**

Page 10, line 31: What do the authors mean by a "broader region"?

Authors' response: We are sorry that it was not clear. We meant "regional transport of air masses", and have changed this in the corresponding text accordingly.

**Comment 16**

Page 11, section 3.2.1: I suggest that the authors provide a table or figure showing timing of onset of nucleation, a simple frequency diagram would suffice. This will show if there is any diurnal preferences regarding onset of NPF.

Authors' response: Thank you for your suggestion. We have provided a Supplementary Figure 1 which shows the frequency of onset of nucleation accordingly.

**Comment 17**

Page 12, line 4-5: It is unclear what the authors mean here. Figure 9 does not show any details about ozone, why it is difficult to follow the authors reasoning. I strongly suggest that the authors provide a more detailed evaluation of the role of O3 in NPF or tracer thereof.

Authors' response:  It was in fact a typo due to our carelessness, which we regret. It meant to refer to Figure 6 instead. The manuscript has been corrected accordingly.

Regarding a more detailed evaluation of the role of O3, we have performed additional analysis as detailed in our response to your Comment 22.

**Comment 18**

Table 3: The numbers do not add up: I assume that data is divided in three main classes: events, non-events and unclassified. Together these should add up to 100%. However, reading the table, March has 83% total, April 92% total, May 112% total. Thus, it needs to be clarified to what the percentage is referring to if not to total amount of observations!?

Authors' response: Thank you for your observant eyes. We apologize for having overlooked the errors in this table. It was due to a formula in the Excel sheet not automatically updated, leading to a wrong total number of days. We have now updated the table to show the right number of days, and thus corrected percentages. Fortunately, the general shares of the events, non-events and undefined are relatively unaffected, as can be seen in the updated table.

We have also added one more significant figure to the numbers shown. Based on what is shown in the table, sometimes the total percentage, for May for example is 99.9%. Here it is simply a mathematical display matter.

We have also updated all the relevant number in text (Section 3.5.2). In addition we have also updated the relevant Figure 9, in which the number of events, non-events and undefined are still the same, while only the total number of days were corrected.

We hope this is now acceptable.

**Comment 19**

Page 12, line 15-16: It seems that the station is within a cloud/fog. Is the inlet whole air or does it have a well defined cut off?

Authors' response: This refers to the sentence "In the late hours of June 19, the $O_3$ concentration suddenly dropped by ~5 ppbv, coinciding with an interruption of the event." Regarding the comment on the inlet, we have provided our response in Comment 5, that the inlet does not have any size cut-off.

According to our new HYSPLIT calculations, this interruption of the event might be related to a change in origin of air masses to a locally confined area. Please see the relevant discussion in Section 3.2.3.

**Comment 20 Henrik**

Page 12, line 24-29: Suggest deleting this paragraph as it has nothing to do with NPF.

Authors' response: We have deleted this paragraph accordingly.

**Comment 21**

Page 13, line 13-15: What does the author mean by "allowing the ozone to recover"? The reasoning seem to assume that the growth control ozone and not vice versa: : : To me, it seems like a change in air-mass.

Authors' response: We agree that the observation could also be due to a change in air-mass. We have therefore removed the sentence.

**Comment 22**

Page 13, line 16-20. This is a quite shaky statement. The variation in ozone and relation formation/growth is not unambiguous. Further, the statement is based on three cases. With less than the authors back the discussion with more statistics, to me the role of ozone seems speculative to say the least. Thus, I strongly suggest that the relation to ozone is studied more in detail. In its current form however, the limited amount of data presented doesn't support anything.

Authors' response: Thank you for your critical comment. We have now calculated the correlation coefficients for all NPF events and O3 available, and have added one more paragraph to describe our observations. We paste this paragraph here:

"The three events seemed to visually display an anti-correlation between, the concentration level of O3 and the growth trend of smaller particlesseemed to display an anti-correlation with early particle growth up to about 30 nm during Event A and Event B or about 40-50 nm in case of Event C. A Pearson correlation coefficient between O3 concentration and integrated particle number concentrations for the nuclei mode range (10-30 nm) was calculated for each event observed during 2012, where O3 data was available, and NOx data was also available to eliminate local pollution spikes. Out of a total of 35 NPF events observed during 2012, 16 events (46% of total events) displayed a weak to moderate anti-correlation (Pearson correlation coefficient below -0.5) between the integrated particle number concentrations for the nuclei mode range (10-30 nm) and O3, with an average coefficient value of -0.71. Meanwhile 12 events (34% of total events) displayed a negative correlation coefficient from -0.05 to -0.41, with an average value of -0.25; and 7 events (20% of total events) showed a positive correlation in the range of 0.09 to 0.44, with an average value of 0.30. In these later cases (54 % of total events), it can be deemed that there is no relationship between O3 and the nucleation mode particle number concentrations. No positive Pearson correlation coefficient stronger than 0.5 was observed."

**Comment 23**

Page 14, line 2-4: The author discuss the role of mass (or rather surface) in regulating amount of small particles via coagulation. Of higher concern would be the role of the condensation sink (CS) of both nucleating and condensing species.

Authors' response: Thank you for your valid comment. This is indeed the case and we have re-written the sentence to emphasize the role of condensation.

**Comment 24**

Page 14, first section: Is the correlation derived for the integral number or for number concentration over some representative size interval? If the latter is not true, I suggest an integration over nuclei mode size range, e.g. 10-30nm, prior attempting correlation tests.

Authors' response: Thank you for your suggestion. We have follow your suggestion adn replied to this comment together with Comment 22. Please see Comment 22 above.

**Comment 25**

Page 14, line 28-31: I do not follow completely. As I read the text, low ozone hinders NO-NO2 transition. So far so good. However, then the author state: "Local NOx emissions thus seemed to relate to: : :". What do the authors mean?? What is the connection?

Authors' response: Thank you for noting this. Since we only report total NOx there should not be any connection here. We have removed the entire paragraph.

**Comment 26**

Page 15, line 8: ": : :at the expense of NOx concentration: : :" I was under the impression that the role of NOx in ozone formation is the photolysis of NO2 to NO+O. Thus, ozone formation does not reduce total levels of NOx.

Authors' response: This is definitely a valid comment. We have revised the sentence accordingly.

**Comment 27**

Page 15, line 26: On how big fraction did this occur? How is the boundary layer defined?

Authors' response: We have revised our entire section on HYSPLIT trajectories, so this sentence is no longer there and the discussion on the boundary layer is no longer the case.

**Comment 28**

Whole discussion using trajectories: The approach using the trajectories is conceptually wrong. All four examples is calculated for trajectory arrival around 01:00 at the end of the event. How does that at all relate to the onset/evolution of the event. In order to be able to discuss anything about the role of the air-mass to some aerosol property, arrival time of trajectory and observation of the parameter must coincide in time. The problem in reasoning is exemplified by e.g. Page 15, lines 28-onwards where the authors discuss mixing along the trajectories. It has to be made clear, that the mixing taking place during 19-june in a trajectory arriving 21-june, 01:00 absolutely has nothing to do with the air arriving to the site 19-june. At least, the author has not shown that such a connection exist.

In order to do the analysis properly, I would recommend the authors to look at the whole time series during the event, calculate one trajectory every hour for the duration of the event. Only then aerosol properties during the event will be linkable to air-mass transport.

As currently used, the conclusions from the trajectory analysis down to page 16, line 26, are completely redundant and should be removed or revised accordingly.

Authors' response: We are very grateful for your critical comment on our trajectory section, which, as you pointed out, indeed requires a re-analysis. We have followed your suggestion and performed hourly backward trajectories for the entire duration of the events. As a result, we have replaced the whole old trajectory discussion with new paragraphs based on the newly calculated trajectories. Accordingly, it seems that NPF events might link to changes in origin of air masses, but not heights. Please see section 3.2.3 for our revision.

**Comment 29**

Page 18, conclusions: The conclusion that ozone is closely related to NPF is not supported by the analysis presented. Three presented cases, and a typically poor correlation between ozone and particle integral number, is clearly not enough. The same goes for the conclusions from the trajectory analysis regarding subsidence. The approach by which the authors derive the conclusion is wrong. Thus, the only verifiable outcome of the analysis is seasonality of NPF events.

Authors' response: We have revised the conclusions accordingly with the results from our new analysis. Thank you again for being straight-forward and critical.

[revised manuscript text omitted]

---

## Author Comment (AC2) · 15 Jun 2016

The manuscript presents an analysis of aerosol characteristics at the high-Arctic site Station Nord in Greenland, based on continuous measurements during 2010–2013 (concentrating on year 2012). The focus of the manuscript is in analysis of new particle formation (NPF) events. Ambient conditions favoring NPF at the site are reported based on case-studies, and NPF events are also analyzed with respect to source areas based on airmass back-trajectories. The dataset presented in the manuscript is interesting, higligting the importance of atmospheric NPF to the aerosol number even in the remote Arctic regions. This work is within the scope of Atmospheric Chemistry and Physics, and could be considered for publication after the comments below have been taken into account.

My major concern is the analysis of the airmasses in relation to the three NPF events presented in the paper. The arrival times of the airmass back-trajectories shown in Figure 7 do not seem to coincide with the NPF events presented in Figure 6. The first airmass shown arrives in-between of the double event A and all the other airmasses of Fig. 7 on the midnight following the events A, B and C. I don't see how any conclusions on aerosol and trace gases measured at Station Nord during these three NPF events could be drawn based on the airmasses shown in Fig 7. Therefore, the analysis of the whole of Section 3.2.3 should be redone by analysing airmasses arriving to the station at the start of the NPF event or at some other relevant time during the event.

Authors' response: We thank you for pointing out our errors in calculating HYSPLIT. We have now re-calculated the backwards trajectories hourly during the entire events, and found that the onset or interruption of the events might be explained by changes of air masses, but not really altitude. We have thereby re-written the entire section on trajectories. Please see Section 3.2.3 for the revision.

Other general comments:

1) Page 2, lines 3–4: "only nucleation and Aitken-mode particles were observed during the summer months". Based on Figures 3–4 and Table 1, this does not seem to be the case. There are clearly particles larger than 100 nm present during all the months, although the concentrations of accumulation mode particles are lower during the summer.

Authors' response: Thank you for your comment. We have revised the sentence to emphasize that nucleation and Aitken-mode particles are the predominant modes, but indeed not the only modes.

2) Also Asmi et al. (2016) reported on NPF observations at the Arctic measurement station Tiksi in northern Siberia. This could be added to the discussion on NPF observations in the Arctic (third paragraph of the Introduction section).

Authors' response: Thank you very much for making us aware of this new publication. We have added Asmi et al. (2016) and relevant discussions to the Introduction section.

3) Page 4, lines 32–33: Is the local pollution source taken into account in the dataanalysis (for example by excluding data when the local wind direction is from the sector towards the pollution source)?

Author's response: The local pollution is mainly from activities in the military camp and the car servicing the station, and sometimes from the airplanes arriving and leaving the station. However there is currently no systematic way of tracking or knowing the exact source/direction of pollution source in combination of wind direction analysis.

So, local pollution is currently deemed as where sudden elevated concentrations of NOx are observed, and thereafter removed from the dataset. This indicator might not cover all types of local emissions that may occur, but at least the major ones.

4) Page 7, line 19: "during the time period from July 2010 to February 2013". I suppose this should be "during 2012", as was stated at the end of Section 2.2.2. Also, Figures 3–5 refer to the year 2012.

Author's response: Actually data from the other years were also used to support the analysis of event statistics. Data from the other years were also used in Figure 8, 9 and Table 3 (together with data from 2012). It should already say on the figure/table captions, but we have added this sentence on page 6 about our use of data from the other years to make it explicit:

"Data from the other years were used to support the analysis of event statistics. Details of the data period used are provided in the caption of the relevant tables or figures."

5) Page 8, line 27: This sentence is little unclear, consider revising to e.g. "Since nucleation mode particles were almost absent in April and relatively minor in May, the high median or average N values observed during these months were attributed to .."

Authors' response: This was indeed a bad sentence. We have removed it completely.

6) Could the analysis of Section 3.2.2 on the role of O3 and NOx in NPF made more general by including data during all the NPF events, instead of just using 3 case studies?

Also, comparison of the O3 and NOx between NPF and non-NPF days could provide useful information. Such analysis should probably be done seasonally in order to exclude the strong difference in NPF occurince between summer and winter.

Authors' response: Thank you for your comment. This is also in line with comment 22 from Referee 1.

To your concern upon the relation between O3 and NPF, we have extended our analysis for all events in 2012, and included more statistics between O3 concentrations and integrated nucleation mode (10-30 nm) particle number concentrations in one additional paragraph in section 3.2.2. We paste the paragraph below.

"The three events seemed to visually display an anti-correlation between, the concentration level of O3 and the growth trend of smaller particles seemed to display an anti-correlation with early particle growth up to about 30 nm during Event A and Event B or about 40-50 nm in case of Event C. A Pearson correlation coefficient between O3 concentration and integrated particle number concentrations for the nuclei mode range (10-30 nm) was calculated for each event observed during 2012, where O3 data was available, and NOx data was also available to eliminate local pollution spikes. Out of a total of 35 NPF events observed during 2012, 16 events (46% of total events) displayed a weak to moderate anti-correlation (Pearson correlation coefficient below -0.5) between the integrated particle number concentrations for the nuclei mode range (10-30 nm) and O3, with an average coefficient value of -0.71. Meanwhile 12 events (34% of total events) displayed a negative correlation coefficient from -0.05 to -0.41, with an average value of -0.25; and 7 events (20% of total events) showed a positive correlation in the range of 0.09 to 0.44, with an average value of 0.30. In these later cases (54 % of total events), it can be deemed that there is no relationship between O3 and the nucleation mode particle number concentrations. No positive Pearson correlation coefficient stronger than 0.5 was observed."

We only mentioned NOx since it has an effect on O3 concentration, and also serves as an indicator of pollution sparks. However it should not have any other direct impacts on NPF events, and therefore we did not include any further analysis on this. We hope this is acceptable.

7) Page 14, lines 29–31: What were the criteria used in the removal of the local pollution episodes? An exceedance of certain NOx level? Why weren't the episodes during August 2nd (Fig. 6) removed from the dataset, if they were identified as local pollution as discussed on lines 19–24 of page 14?

Authors' response: We defined local pollution episodes in Section 2.2.1. We have tried to re-write this part a bit as follows to make it clearer: "Subsequently, daily particle number size distributions were plotted to inspect any sudden increase in the particle number concentration above the background. If such sudden increase in particle number concentration peaked (without any detectable particle growth) coincided with sudden elevation of NOx concentration, they were interpreted as local pollution events and excluded from the data set."

The episodes on August 2$^{nd}$ were meant as "examples". We have now added an explicit sentence (in Section 3.2.2, NOx part) to explain that they were not used for data analysis. "Such episodes with NOx interference are also demonstrated here as example and were not included in any calculations of data".

8) Page 15, lines 26–27: Are you certain that the airmasses descend from above the boundary layer in these two cases? At least for Event C the airmass arriving at 50 mheight stays constantly below 250 m, which seems quite low to be above the boundary layer.

Authors' response: We have redone the HYSPLIT analysis, so this point no longer holds. Please see section 3.2.3 for our new analysis.

9) Page 15, lines 28–30: I don't fully understand how can the vertical mixing of the airmasses be inferred from Fig. 7 for the case of mid-day of June 19. According to the map, the two airparcels do not follow the same horizontal path, so even though they are at the same altitude at the same time on mid-day of June 19, they are not at the same location horizontally and therefore not interacting with each other.

Authors' response: This was indeed our mistake. We have redone the HYSPLIT analysis, so again this point no longer holds. Please see section 3.2.3 for the revised section instead.

10) Page 17, lines 1–2: Is the map of Figure 8 constructed using all the trajectories arriving at Station Nord during the year 2012? Is the number of trajectories big enough for drawing conclusions on the source areas of airmasses favouring NPF?

Authors' response: Thank you for your comment. We have added this additional sentence to the caption of Figure 8: "This figure uses all available data (62 events) from the study period July 2010 – February 2013." This is definitely not an overwhelming number, but represents our current best available data at the station to date. We hope providing this extra information on the size of the number of events would help the readers to judge the reliability of the map.

11) Page 17, lines 30–31: Asmi et al. (2016) reported similar NPF day frequency, 30–40%, during summer in Tiksi, Russia.

Authors' response: Once again, we thank you for providing us with this interesting paper. We have added a few sentences to the discussion highlighting the similarities and differences in our observations and Asmi et al. (2016).

12) In the conclusions, the statement on the close relationship of ozone to the particle growth (lines 18–19) seems hard to justify on the basis of the presented material, which is currently 3 case studies of NPF. What were the exact growth rates of 10–25 nm particles during these events? Could this analysis be made more thorough by including all the NPF days and showing the relationship between O3 concentration and particle growth rates (see also my comment 6)?

Authors' response: Thank you for your suggestion. To perform a statistical analysis on ozone vs growth rate during all the new particle formation events is a very good idea and deserves attention in a manuscript focusing entirely on new particle formation. However, it was not the intention to completely dominate the focus on the new particle formation events in this paper. So, such an analysis would take too much space in this paper and we feel that it might be out of the current scope.

Technical comments:

Page 1, line 23: "focus" should be "focuses"

Authors' response: Thank you for noting this. It has been corrected accordingly.

Page 6, line 16: section number "2.2.2" should be "2.2.3"

Authors' response: Definitely! It has been corrected.

Page 12, line 5: "Fig. 9" should be "Fig. 6"

Authors' response: We have corrected the figure number accordingly

Page 17, line 30–31: ".. relatively higher compared to .." should be "relatively high compared to .."

Authors' response: Thank you for your suggestion. We have revised the sentence accordingly.

References:

[revised manuscript text omitted]

---

## Author Comment (AC3) · 15 Jun 2016

Dear reviewers,

We would like to thank both reviewers for having taken your time to read our manuscript so thoroughly. We also highly appreciate your critics, comments and suggestions. We have addressed the points raised, and we believe that the manuscript has improved considerably owing to taking into account your comments.

Please see our attached file for our reply and revised manuscript.

[Figure]

Yours sincerely,

Quynh Nguyen on behalf of the authors
* * *

---

## Referee Report (RR1)

Review of the revised version of the manuscript

"Seasonal variation of atmospheric particle number concentrations, new particle formation and atmospheric oxidation capacity at the high Arctic site Villum Research Station, Station Nord"

by Nguyen et al., submitted to Atmospheric Chemistry and Physics (acp-2016-205)

The authors have responded adequately to the issues raised in my and the other referees comments. There still remains one additional point to clarify (I have discussed about this also with the editor of the manuscript, Dr. M. Boy): in Section 2.2.3 (page 7 of the revised manuscript) it is said that the detection limit of the NOx monitor was 150 ppt. However, in Figure 6 almost all the NO and NOx values shown are below 0.1 ppb. Also in the Results section in several places NOx concentrations below 0.1 ppb are discussed, without any mention that these concentrations are below the detection level of the NOx monitor. While I understand that this issue will not change the conclusions of the manuscript, could the authors still clarify in the manuscript how reliable is the NOx data in the very low concentration conditions of the Station Nord?

After clarifying the NOx data issue the manuscript can be accepted for publication in Atmospheric Chemistry and Physics.